

# The effects of viewing cute images on the performance of simple computerized tasks in dog owners and non-dog owners

Orly Fox and Gal Ziv

The Academic College at Wingate, Netanya, Israel

## ABSTRACT

**Background:** Viewing cute images of puppies or kittens can improve the performance of various motor tasks due to increased behavioral carefulness. It is possible, however, that this effect can be moderated by the affinity of individuals towards specific cute stimuli. The purpose of this pre-registered study was to examine whether dog ownership moderates the effect of viewing cute images of puppies on two computerized RT-based tasks.

**Methods:** One-hundred and sixty-four participants were covertly recruited and randomly assigned to four groups: (1) dog owners viewing puppies' images ($n = 35$), (2) dog owners viewing adult dogs' images ($n = 36$), (3) non-dog owners viewing puppies' images ($n = 48$), and (4) non-dog owners viewing adult dogs' images ($n = 45$). The participants performed a Simon task and an alternate task-switching task before and after viewing seven images of puppies/adult dogs based on group affiliation. After performing the tasks, the participants rated each image on five characteristics: cuteness, infantility, pleasantness, excitability, and wanting to get closer.

**Results:** The participants who viewed images of puppies rated those characteristics (*e.g.*, cuter, more infantile, *etc.*) as higher than the participants who viewed images of adult dogs. There were no differences in the performance of the two tasks between participants who viewed images puppies and participants who viewed images of adult dogs. However, dog owners reacted faster than non-dog owners in the post-test of the alternate task-switching task. It is possible that showing images of dogs to dog owners increased their motivation and attention to the task.

# INTRODUCTION

Humans appear to engage in caregiving behaviors (*i.e.*, behavioral carefulness–behaving carefully to avoid causing harm) when they face cute or infantile appearances (*Sherman, Haidt & Coan, 2009*). Eliciting such carefulness can be useful when individuals are required to perform tasks without making errors (*e.g.*, caring for the young or engaging in dangerous tasks such as operating heavy machinery). But how does one elicit behavioral carefulness without the presence of the cute organism?

It is possible, for example, to present images of cute and/or infantile animals or humans to individuals. Indeed, several studies have compared the performance of various tasks

Corresponding author
Gal Ziv, galziv@wincol.ac.il

between participants who viewed images of adult animals and participants who viewed images of young/baby animals or infants (*Álvarez-San Millán et al., 2021*; *Nittono et al., 2012*; *Sherman, Haidt & Coan, 2009*; *Yoshikawa & Masaki, 2021*; *Yoshikawa, Nittono & Masaki, 2020*; *Ziv & Fox, 2021*). Most of these studies found that viewing cute images affects performance in various tasks. Specifically, this effect was found when participants performed dexterity tasks (*Nittono et al., 2012*, Exp. 1; *Sherman, Haidt & Coan, 2009*; *Yoshikawa, Nittono & Masaki, 2020*), sport-related tasks (*e.g.*, basketball free throw; *Yoshikawa & Masaki, 2021*), reaction time (RT) tasks; (*Nittono et al., 2012*, Exp. 3), or visual processing/attentional tasks (*Álvarez-San Millán et al., 2021*). For example, in one experiment that used a manual dexterity task (*Nittono et al., 2012*, Exp. 1), after viewing images of baby animals, performance accuracy improved while the time to complete a task increased. In another experiment from the same article (*Nittono et al., 2012*, Exp. 2), after viewing images of baby animals the number of correct responses in a visual search task increased. However, *Álvarez-San Millán et al. (2021)* reported slower RTs in a global visual search task after viewing cute images, but not in a local search task.

In contrast to the abovementioned studies, a recent study failed to find differences in the performance of computerized RT-based tasks between participants who viewed images of young pets and participants who viewed images of adult pets (*Ziv & Fox, 2021*). However, when the authors compared pet owners to non-pet owners, the effect of viewing cute images emerged in pet owners only. Therefore, it is possible that pet ownership moderated the effects of behavioral carefulness on performance. Compared to non-pet owners, pet owners might be prone to perceiving pets in general, and young pets in particular, as cuter, and the effects of viewing those images on their affect might be greater. This positive affect, in turn, may increase their motivation and their attention to the task. It is also noteworthy that in all previous studies, the authors used images of both dogs and cats, and in some instances other animals (*e.g.*, *Sherman, Haidt & Coan, 2009*). The use of different animals can be another moderating variable due to possible general affinity of individuals to one species.

Our purpose, in the current study, was two-fold. First, we wanted to avoid the possible moderating effect of the affinity towards different species (*e.g.*, cat *vs* dog) and therefore, we used images of dogs only. Second, we attempted to directly examine whether there are differences between dog owners and non-dog owners in their performance of computerized RT-based tasks after viewing images of puppies or adult dogs. To do that, we conducted an online study and recruited participants without them knowing that the study had anything to do with the fact that they are, or they are not dog owners. We chose two tasks that require attention—a Simon task and an alternate task-switching task. In a Simon task, half of the trials have incongruence between the location of the cue and the meaning of the cue (*e.g.*, *Lu & Proctor, 1995*). In an alternate task-switching task, a specific task rule changes when the stimuli are presented in different locations. This change requires participants to attend to a different attribute of the stimuli (*Monsell, 2003*). We chose these tasks because we wanted to directly examine whether we can replicate the finding in the exploratory analysis of *Ziv & Fox (2021)*, that the behavioral carefulness effect was found in pet owners only. Therefore, we used the exact same tasks as *Ziv & Fox (2021)*. In addition,
**Table 1 Ratings of the seven images of adult dogs/cats and the seven images of puppies/kittens that were chosen for the main study (means ± SD).**

| Attribute | Dog ownership | Adult dogs | Puppies | Statistics |
|---|---|---|---|---|
| Cuteness | Owner ($n$ = 12) | 3.15 ± 1.17 | 4.52 ± 0.81 | $t(11)$ = 3.91, $p$ = 0.002 |
| | Non-owner ($n$ = 9) | 1.71 ± 1.58 | 4.51 ± 0.34 | $t(8)$ = 5.54, $p$ = 0.001 |
| | Both ($n$ = 21) | 2.5 ± 1.5 | 4.5 ± 0.6 | $t(20)$ = 6.1, $p$ < 0.001 |
| Infantility | Owner ($n$ = 12) | 1.33 ± 0.65 | 4.30 ± 0.68 | $t(11)$ = 10.72, $p$ < 0.001 |
| | Non-owner ($n$ = 9) | 0.71 ± 1.09 | 4.08 ± 0.64 | $t(8)$ = 9.36, $p$ < 0.001 |
| | Both ($n$ = 21) | 1.1 ± 0.9 | 4.2 ± 0.7 | $t(20)$ = 14.3, $p$ < 0.001 |
| Pleasantness | Owner ($n$ = 12) | 2.61 ± 0.87 | 4.45 ± 0.63 | $t(11)$ = 6.27, $p$ < 0.001 |
| | Non-owner ($n$ = 9) | 1.73 ± 1.62 | 4.43 ± 0.38 | $t(8)$ = 4.87, $p$ = 0.001 |
| | Both ($n$ = 21) | 2.2 ± 1.3 | 4.4 ± 0.5 | $t(20)$ = 7.4, $p$ < 0.001 |
| Excitability | Owner ($n$ = 12) | 2.44 ± 1.13 | 4.10 ± 0.92 | $t(11)$ = 4.78, $p$ = 0.001 |
| | Non-owner ($n$ = 9) | 2.27 ± 1.58 | 3.94 ± 0.83 | $t(8)$ = 2.72, $p$ = 0.026 |
| | Both ($n$ = 21) | 2.4 ± 1.3 | 4.0 ± 0.9 | $t(20)$ = 5.2, $p$ < 0.001 |
| Wanting to get closer | Owner ($n$ = 12) | 2.98 ± 1.47 | 4.60 ± 0.73 | $t(11)$ = 3.52, $p$ = 0.005 |
| | Non-owner ($n$ = 9) | 2.24 ± 1.73 | 4.38 ± 0.53 | $t(8)$ = 3.68, $p$ = 0.006 |
| | Both ($n$ = 21) | 2.7 ± 1.6 | 4.5 ± 0.6 | $t(20)$ = 5.2, $p$ < 0.001 |

to prevent the possible moderating effects of type of pets, we used only images of puppies and adult dogs and only dog owners and non-dog owners. In addition, such attentional tasks should be suitable for examining the effects on behavioral carefulness because to behave carefully and mindfully, attention is required. Finally, such tasks can be easily implemented in an online study. We hypothesized that: (1) In dog owners, in both a Simon and an alternate task-switching task, participants who viewed images of puppies would make more correct responses and react faster compared with participants who viewed images of adult dogs, and (2) in non-dog owners, these effects would not be found.

## MATERIALS AND METHODS

### Pre-registration and raw data repository

The study was pre-registered on aspredicted.org (https://aspredicted.org/eu4ze.pdf). Analyses that were not pre-registered are reported as "exploratory analyses". The raw dataset is available on OSF (https://osf.io/48azq/?view_only=1326169e9fcc4989945ef4f5ccd78ad3).

### Preliminary image selection

The images for the study were chosen from a preliminary study that asked 21 participants to rate 15 images of puppies and 15 images of adult dogs on a scale of 0 to 5 for the following criteria: cuteness, infantility, pleasantness, excitement, wanting to get closer. Out of those, seven images of puppies and seven images of adult dogs were chosen for the main study. Images were chosen to produce large differences in the rating. Table 1 shows the differences in ratings between adult dogs and puppies. Note that although we

pre-registered 30 participants for this stage, after 21 participants rated the images, it was clear which images can be chosen and so we stopped recruitment of participants for this stage.

## Main study

### Participants

We used simulation-based power analysis (*Lakens & Caldwell, 2021*) to find a sample size that will allow us to reach 80% statistical power to find a three-way interaction (Image group (puppies/adult dog images) X Dog ownership (yes/no) X Test (pre/post)) in the alternate task-switching task. For the simulation, we used data from a previous study that we conducted on the same topic (*Ziv & Fox, 2021*). The simulation showed that the expected effect size for the three-way interaction was $\eta^2_p = 0.05$ and that 39 participants per group (156 participants in total) were required to achieve 80% power. Therefore, we attempted to recruit 180 participants to account for possible dropouts. In actuality, One-hundred and seventy-eight participants completed the study. The participants were recruited from prolific.co—an online participants' database. Out of those participants, 164 participants (mean age = 26.06 ± 4.65 years, 114 females) were included in the final analysis (see Data Exclusion section). The participants were randomly assigned to four groups: (1) dog owners viewing images of puppies (*n* = 35, 24 females), (2) dog owners viewing images of adult dogs (*n* = 36, 28 females), (3) non-dog owners viewing images of puppies (*n* = 48, 33 females), and (4) non-dog owners viewing images of adult dogs (*n* = 45, 29 females). Recruitment to the study was covert based on information the participants provided when they joined prolific.co. To make sure that this information was valid, we also asked, at the end of the study, whether the participants owned a dog. There were some discrepancies between the answers to the study's questionnaire and the information available on prolific.co. When such discrepancies occurred, we relied on the more recent answers to the study's questionnaire and therefore, the groups are unequal. Randomization was accomplished automatically by the web-based platform used for the study (Gorilla.sc).

The participants in the preliminary image selection stage were not paid but the participants in the main study were paid £2.5 for their participation. The study was approved by the ethics committee of the Academic College at Wingate (approval # 315), and all participants signed an informed consent form (in the preliminary image selection stage) or completed an electronic informed consent form on the study's website prior to participation (for the main study).

### Tasks

#### Simon task

The word "right" or the word "left" was displayed on the right or the left side of a central cross. Participants were asked to press the key "j" when they saw "right" (even if it appeared on the left side of the cross) and to press the key "f" when they saw "left" (even if it appeared on the right side of the cross) (*Lu & Proctor, 1995*; *Simon & Wolf, 1963*).

The words "left" or "right" were shown for 900 msec, followed by 600 msec during which only a central cross was shown.

*Alternate task-switching task*

A rectangle or square, in a green or blue color, were shown on the upper half or the lower half of the screen. When the shape appeared at the upper half, the participants had to press the key "j" if the shape was green and the key "f" if the shape was blue (irrespective of the shape itself); If the shape appeared at the bottom half, the participants had to press "j" if the shape was a rectangle and "f" if it was a square (irrespective of the color). The shape was displayed for an unlimited time until a key press was recorded. Two shapes were presented at the upper half or at the bottom half of the screen, alternately.

## Procedure

The participants performed three blocks of 24 trials in each of the two tasks in a pre-test and in a post-test. Generally, each block took less than 30 or 45 s to complete the Simon task or the alternate task-switching task, respectively. Between pre- and post-tests, participants viewed seven images of either puppies or adult dogs based on group affiliation and were asked to rate them based on their preference, from the image they liked the most to the image they liked the least. We used seven images because previous studies that used the same number of images showed an effect on performance (*e.g.*, *Nittono et al., 2012*; *Yoshikawa, Nittono & Masaki, 2020*; *Yoshikawa & Masaki, 2021*). After the post-test, participants were asked to rate the images on a scale of 0 (not at all) to 5 (very much) for the following criteria: cuteness, infantility, pleasantness, excitement, wanting to get closer. Finally, all participants were asked three questions: (1) how much they loved dogs on a scale of 0 (not at all) to 5 (very much), (2) do they have a dog (yes/no), and (3) do they have another pet (yes/no).

## Data exclusion

In accordance with our pre-registration, in the Simon task, we excluded RT values of over 1,000 msec because they were longer than duration of stimulus presentation (900 msec). We also excluded blocks with 13 or less correct responses. These blocks were deleted because they represented less than 55% correct responses suggesting that the participants were not attentive to the task or motivated to perform it according to the instructions (for example, 50% correct responses can be achieved simply by pressing the same key continuously). This led to the removal of three blocks in the pre-test and eight blocks in the post-test. Similarly, in the alternate task-switching task, we excluded blocks with 13 or less correct responses. This led to the removal of 42 blocks from the pre-test and 22 blocks from the post-test. Fourteen participants were removed from the study because they had more than two blocks that had to be removed from one of the two tasks. These exclusions criteria were pre-registered.

## Data analyses

Normality was assessed using Kurtosis and Skewness values (values under 2 were considered acceptable). Skewness values were mostly acceptable but in several cases kurtosis values were not. Still, we used parametric statistics because it has been suggested
that analysis of variance (ANOVA) is robust to such violations and that type I and type II errors remain constant under different distributions (*Schmider et al., 2010*). We conducted three-way ANOVAs (Image group (puppies/adult dogs' images) X Dog ownership group (yes/no) X Test (pre/post)) with repeated measures on the Test factor to assess differences in reaction times and number of correct key presses. To assess the interactions in switch/no-switch trials we conducted two three-way ANOVAs for the pre-test and the post-test: Image group X Dog ownership group X Switch (yes/no). Bonferroni post-hoc analyses and 95% confidence intervals were used for post-hoc testing when necessary. To assess differences in ratings of images' attributes we used a two-way ANOVA (Image group X Dog ownership Group). In cases where the $p$ value was over 0.05 but under 0.10, and at the same time the effect size was moderate or above (Cohen's d $\geq$ 0.5 or $\eta^2_p \geq 0.06$), we considered this finding as practically relevant and discussed it as such.

In our exploratory analyses we used Bayesian analyses to support our pre-registered ANOVAs. We also conducted two-way ANOVAs to examine the relationship between the love for dogs and performance. All analyses were performed using JASP version 0.16.2 (*JASP Team, 2022*) and SPSS 25 (IBM Corp, Armonk, NY, USA). Statistical significance level was set at 0.05.

## RESULTS

### Simon task

#### *RT*

A three-way ANOVA (Image group X Dog ownership group X Test) with repeated measures on the Test factor revealed a Test effect, $F(1, 160) = 13.08$, $p < 0.001$, $\eta^2_p = 0.08$. The mean RT was 508.79 ± 79.82 msec in the pre-test and 497.83 ± 69.87 msec in the post-test. There was no Image group effect, $F(1, 160) = 0.11$, $p = 0.740$, $\eta^2_p = 0.00$, no Dog ownership group effect, $F(1, 160) = 1.31$, $p = 0.254$, $\eta^2_p = 0.01$, no Test X Image group interaction, $F(1, 160) = 2.13$, $p = 0.146$, $\eta^2_p = 0.01$, no Test X Dog ownership group interaction, $F(1, 160) = 0.08$, $p = 0.771$, $\eta^2_p = 0.00$, no Image group X Dog ownership group interaction, $F(1, 160) = 0.04$, $p = 0.843$, $\eta^2_p = 0.00$, and no three-way interaction, $F(1, 160) = 2.13$, $p = 0.147$, $\eta^2_p = 0.01$.

#### *Correct responses*

There were no significant main effects or interactions (all $p$ values > 0.148, all $\eta^2_p <= 0.01$). The mean correct responses was 22.50 ± 1.07 and 22.54 ± 1.03 in the pre-test and the post-test, respectively (out of 24).

### Alternate task-switching task

#### *RT*

A three-way ANOVA (Image group X Dog ownership group X Test) with repeated measures on the Test factor revealed a Test effect, $F(1, 160) = 118.01$, $p < 0.001$, $\eta^2_p = 0.42$. The mean RT was 1,046.19 ± 280.05 msec in the pre-test and 896.31 ± 205.90 msec in the post-test. There was no Image group effect, $F(1, 160) = 0.10$, $p = 0.751$, $\eta^2_p = 0.00$, no Dog ownership group effect, $F(1, 160) = 1.74$, $p = 0.189$, $\eta^2_p = 0.01$, no Image group X Test

interaction, $F(1, 160) = 0.65$, $p = 0.420$, $\eta^2_p = 0.00$, no Dog ownership group X Test interaction, $F(1, 160) = 1.67$, $p = 0.198$, $\eta^2_p = 0.01$, no Image group X Dog ownership group interaction, $F(1, 160) = 0.37$, $p = 0.546$, $\eta^2_p = 0.00$, and no three-way interaction, $F(1, 160) = 0.02$, $p = 0.882$, $\eta^2_p = 0.00$.

### Correct responses

There were no significant main effects or interactions (all $p$ values > 0.154, all $\eta^2_p <= 0.02$). The mean correct responses was 21.41 ± 2.54 and 22.21 ± 1.77 in the pre-test and the post-test, respectively (out of 24).

## Analyses of switch and no-switch trials

### Pre-test RT

A three-way ANOVA (Image group X Dog ownership group X Switch condition) with repeated measures on the Switch condition factor revealed a Switch condition effect, $F(1, 160) = 290.68$, $p < 0.001$, $\eta^2_p = 0.64$. The mean RT was 1,177.97 ± 302.29 msec in the switch trials and 914.41 ± 290.63 msec in the no-switch trials. There was no Image group effect, $F(1, 160) = 0.26$, $p = 0.608$, $\eta^2_p = 0.00$. There was also no Dog ownership group effect, $F(1, 160) = 0.44$, $p = 0.506$, $\eta^2_p = 0.01$. The mean RT of the dog owners (1,029.84 ± 294.04 msec) was similar to the mean RT of the non-dog owners (1,058.67 ± 269.82 msec) (Fig. 1). In addition, there was no Switch condition X Image group interaction, $F(1, 160) = 0.21$, $p = 0.647$, $\eta^2_p = 0.00$, no Switch condition X Dog ownership group interaction, $F(1, 160) = 0.18$, $p = 0.668$, $\eta^2_p = 0.00$, no Image group X Dog ownership group interaction, $F(1, 160) = 0.29$, $p = 0.589$, $\eta^2_p = 0.00$, and no three-way interaction, $F(1, 160) = 0.04$, $p = 0.839$, $\eta^2_p = 0.00$.

### Pre-test correct responses

A three-way ANOVA (Image group X Dog ownership group X Switch condition) with repeated measures on the Switch condition factor revealed a Switch condition effect, $F(1, 160) = 4.13$, $p = 0.044$, $\eta^2_p = 0.03$. The mean correct responses was 10.76 ± 1.27 in the switch trials and 10.64 ± 1.41 in the no-switch trials. There was no Image group effect, $F(1, 160) = 1.02$, $p = 0.314$, $\eta^2_p = 0.01$, no Dog ownership group effect, $F(1, 160) = 1.95$, $p = 0.165$, $\eta^2_p = 0.01$, no Switch condition X Image group interaction, $F(1, 160) = 0.06$, $p = 0.805$, $\eta^2_p = 0.00$, no Switch condition X Dog ownership group interaction, $F(1, 160) = 1.55$, $p = 0.215$, $\eta^2_p = 0.01$, no Image group X Dog ownership group interaction, $F(1, 160) = 0.09$, $p = 0.761$, $\eta^2_p = 0.00$, and no three-way interaction, $F(1, 160) = 0.11$, $p = 0.738$, $\eta^2_p = 0.00$.

### Post-test RT

A three-way ANOVA (Image group X Dog ownership group X Switch condition) with repeated measures on the Switch condition factor revealed a Dog ownership group effect, $F(1, 160) = 4.15$, $p = 0.043$, $\eta^2_p = 0.03$. The mean RT of the dog owners (859.08 ± 190.18 msec) was faster than the mean RT of the non-dog owners (924.74 ± 213.79 msec) (Fig. 1). There was also a Switch condition effect, $F(1, 160) = 311.68$, $p < 0.001$, $\eta^2_p = 0.66$. The mean RT was 1,035.12 ± 279.13 msec in the switch trials and 808.66 ± 258.42 msec in
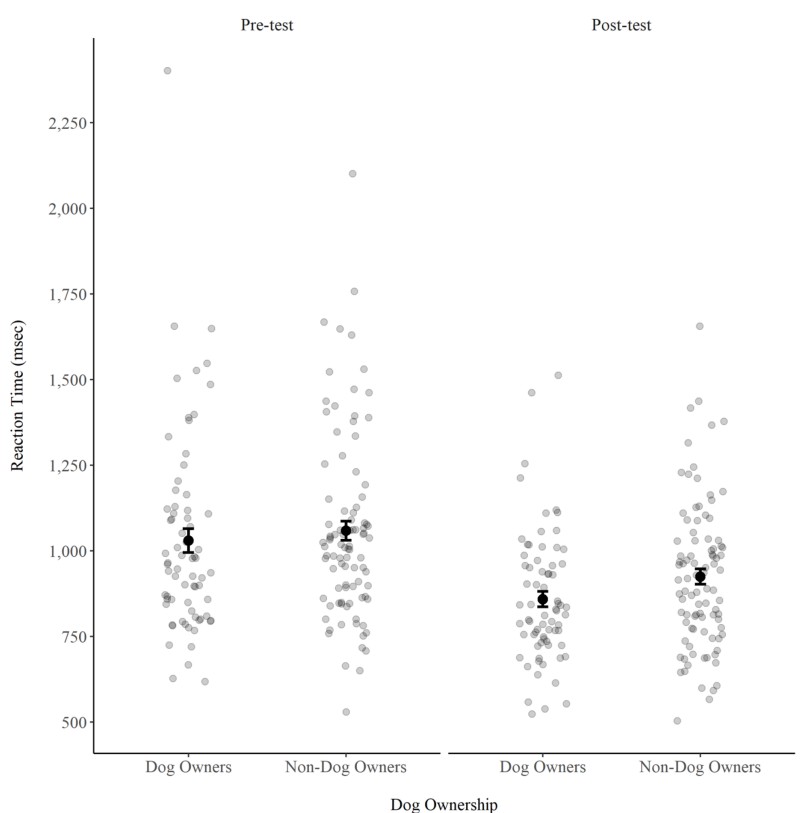

**Figure 1** **Reaction time (msec) in the alternate task-switching task pre-test and post-test for both dog owners and non-dog owners.** These data are from the three-way ANOVA analysis that compared switch and no-switch trials. Error bars represent standard error.

the no-switch trials. There was no Image group effect, $F(1, 160) = 0.00$, $p = 0.996$, $\eta^2_p = 0.00$, no Switch condition X Image group interaction, $F(1, 160) = 0.13$, $p = 0.718$, $\eta^2_p = 0.00$, no Switch condition X Dog ownership group interaction, $F(1, 160) = 1.48$, $p = 0.225$, $\eta^2_p = 0.01$, no Image group X Dog ownership group interaction, $F(1, 160) = 0.38$, $p = 0.540$, $\eta^2_p = 0.00$, and no three-way interaction, $F(1, 160) = 0.34$, $p = 0.560$, $\eta^2_p = 0.00$.

### Post-test correct responses

There were no significant main effects or interactions (all $p$ values > 0.282, all $\eta^2_P <= 0.01$). The mean correct responses was 11.14 ± 1.00 and 11.07 ± 0.94 in the switch trials and the no-switch, respectively (out of 12).

### Rating of images' attributes

Table 2 presents descriptive and inferential statistics of the two-way ANOVAs (Image group X Dog ownership group) that were conducted to examine the differences in the ratings of the five image attributes: cuteness, infantility, excitement, pleasantness, and wanting to get closer. Differences between participants who viewed images of puppies and those who viewed images of adult dogs were statistically significant in all five attributes. Differences between dog owners and non-dog owners were statistically significant only for cuteness and pleasantness. No interactions were found.

**Table 2 Rating of images' attributes (mean ± SD).** Differences in bold are statistically significant.

| Attribute | Factor | | Mean ± SD | F (df) | p | $\eta^2_p$ |
|---|---|---|---|---|---|---|
| Cuteness | Image group | Puppies | **4.20 ± 0.85** | 136.48 (1, 160) | <0.001 | 0.46 |
| | | Adult dogs | **2.71 ± 0.81** | | | |
| | Dog ownership group | owner | **3.59 ± 1.05** | 3.98 (1, 160) | 0.048 | 0.02 |
| | | Non-owner | **3.37 ± 1.16** | | | |
| | Interaction | | | 0.93 (1, 160) | 0.338 | 0.01 |
| Infantility | Image group | Puppies | **3.55 ± 1.01** | 161.82 (1, 159) | <0.001 | 0.50 |
| | | Adult dogs | **1.69 ± 0.88** | | | |
| | Dog ownership group | owner | 2.73 ± 1.45 | 2.59 (1, 159) | 0.110 | 0.02 |
| | | Non-owner | 2.55 ± 1.23 | | | |
| | Interaction | | | 2.02 (1, 159) | 0.157 | 0.01 |
| Excitement | Image group | Puppies | **3.25 ± 1.04** | 59.68 (1, 160) | <0.001 | 0.27 |
| | | Adult dogs | **2.10 ± 0.86** | | | |
| | Dog ownership group | owner | 2.73 ± 1.18 | 0.61 (1, 160) | 0.436 | 0.00 |
| | | Non-owner | 2.64 ± 1.06 | | | |
| | Interaction | | | 0.75 (1, 160) | 0.388 | 0.00 |
| Pleasantness | Image group | Puppies | **4.05 ± 0.86** | 181.74 (1, 160) | <0.001 | 0.53 |
| | | Adult dogs | **2.37 ± 0.75** | | | |
| | Dog ownership group | owner | **3.42 ± 1.03** | 9.63 (1, 160) | 0.002 | 0.06 |
| | | Non-owner | **3.07 ± 1.24** | | | |
| | Interaction | | | 0.80 (1, 160) | 0.372 | 0.00 |
| Wanting to get closer | Image group | Puppies | **3.75 ± 1.12** | 74.07 (1, 160) | <0.001 | 0.32 |
| | | Adult dogs | **2.34 ± 0.95** | | | |
| | Dog ownership group | owner | 3.11 ± 1.25 | 0.68 (1, 160) | 0.411 | 0.00 |
| | | Non-owner | 3.01 ± 1.27 | | | |
| | Interaction | | | 0.001 (1, 160) | 0.974 | 0.00 |

## Time participants spent rating dogs' images

This analysis was conducted to make sure there was no difference in the time it took participants to look at the seven images of puppies or adult dogs and to rank them from the image they liked the most to the image they liked the least. A two-way ANOVA [Image group (young/adult) X Dog ownership group (owner/non-owner)] revealed no Image group effect, $F(1, 160) = 1.20$, $p = 0.274$, $\eta^2_p = 0.01$, no Dog ownership effect, $F(1, 160) = 0.11$, $p = 0.744$, $\eta^2_p = 0.00$, and no interaction, $F(1, 160) = 0.33$, $p = 0.568$, $\eta^2_p = 0.00$. The mean time participants spent rating the seven images according to their preferences was 45.39 ± 22.17 s for those who viewed puppies and 42.09 ± 18.10 s for those who viewed adult dogs. Similarly, the mean time spent rating preferred images was 43.13 ± 21.94 s in dog owners and 44.24 ± 19.00 s in non-dog owners.

## Exploratory analyses
### *Elaborating on differences between dog owners and non-dog owners*

In our pre-registered analysis of the post-test, the only significant finding was that dog owners reacted faster than non-dog owners in the alternate task-switching task (in the analysis of the Switch and No-Switch Trials). To assess whether this may have been a

**Table 3 Number of dog owners and non-dog owners in each category of the dog loving scale.**

| Dog loving scale | 0 | 1 | 2 | 3 | 4 | 5 | Total |
|---|---|---|---|---|---|---|---|
| Dog owners | 0 | 0 | 1 | 3 | 10 | 55 | 69 |
| Non-dog owners | 1 | 9 | 7 | 24 | 23 | 29 | 93 |
| Total | 1 | 9 | 8 | 27 | 33 | 84 | 162 |

Note:
Dog loving scale: 0 (not at all) – 5 (very much).

spurious finding, we examined this in two alternative methods. First, instead of examining the pre-test and the post-test separately, we conducted a four-way ANOVA (Image group X Dog ownership group X Switch condition X Test) with repeated measures on the two latter factors. There was an expected Switch effect, $F(1, 160) = 410.09$, $p < 0.001$, $\eta^2_p = 0.72$, and a Test effect, $F(1, 160) = 118.01$, $p < 0.001$, $\eta^2_p = 0.42$. However, there were no other significant findings (all $p$ values > 0.119). Specifically, unlike the pre-registered analysis, there was no Dog ownership effect, $F(1, 160) = 1.74$, $p = 0.189$, $\eta^2_p = 0.01$.

Second, we conducted a Bayesian three-way ANOVA for the post-test RT in the alternate task-switching task. This analysis showed that the best model to explain the post-test RT of the alternate task-switching task included the Switch factor and the Dog ownership factor. Compared to this best model (which receives the default value $BF_{10} = 1$ in the analysis), the Switch factor alone was second in likelihood ($BF_{10} = 0.64$). $BF_{10}$ values for all other models were under 0.34 (for the null model, $BF_{10}$ was 4.43E-38). Alternatively, when conducting the Bayesian analysis in comparison to the null model (which receives the default value $BF_{10} = 1$), the $BF_{10}$ for the model that included the Switch factor and the Dog ownership factor was highest (2.26E+37).

### Dog loving score and performance

Table 3 presents the distribution of participants across the dog loving scale (0–5). As expected, most dog owners (80%) marked 5 on this scale. Because there were only 18 participants that marked 0, 1, or 2, we collapsed these categories with those who marked 3 and created three groups: those who marked 0–3 ($n = 45$), those who marked 4 ($n = 33$), and those who marked 5 ($n = 84$). We then conducted two-way ANOVAs [Dog loving group X Test] to examine differences in performance variables. There were no significant findings for correct responses in either the Simon task or the alternate task-switching task.

However, for the RTs in the Simon task, the analysis revealed a Dog loving group effect, $F(2, 159) = 5.35$, $p = 0.006$, $\eta^2_p = 0.06$. A Bonferroni post-hoc analysis revealed that participants who marked 0–3 were slower (532.90 ± 87.69 msec) than participants who marked 4 (489.62 ± 54.95 msec) and participants who marked 5 (493.36 ± 66.35 msec). There was also a Test effect, $F(1, 159) = 10.67$, $p = 0.001$, $\eta^2_p = 0.06$, but no interaction, $F(2, 159) = 2.97$, $p = 0.054$, $\eta^2_p = 0.04$.

For the RTs in the alternate task-switching task, the same analysis also revealed a Dog loving group effect, $F(2, 159) = 6.53$, $p = 0.002$, $\eta^2_p = 0.08$. A Bonferroni post-hoc analysis revealed that participants who marked 0–3 were slower (1,073.69 ± 276.30 msec) than participants who marked 5 (927.14 ± 192.67 msec) but not from participants who marked

4 (954.45 ± 209.44 msec). There was also a Test effect, $F(1, 159) = 91.60$, $p < 0.001$, $\eta^2_p = 0.37$, but no interaction, $F(2, 159) = 0.29$, $p = 0.750$, $\eta^2_p = 0.00$.

### Bayesian analyses

We conducted three-way Bayesian ANOVAs (Image group X Dog ownership group X Test) to assess the probability of all models compared to the best model. For the Simon RT, the best model only included the Test factor. $BF_{01}$ was 1.68 for a model that included the Test factor and the Dog owner group factor; 2.54 for a model that included all three main factors; and >4.20 for all other possible models (with most models >10). Similar findings were found for the RTs and correct responses in the alternate task-switching task. For the correct responses in the Simon task, the null model was the best. $BF_{01}$ was 3.61 for the Dog owner group factor; 3.96 for the Image group factor, and >7.46 for all other factors (with most models >10).

The results of the Bayesian analyses strengthen our pre-registered analyses. Bayes Factors mostly provided moderate to strong evidence that models that include anything other than the Test factor were less likely than the model that includes the Test factor alone. However, there was some likelihood for models that include the Dog owner group as well. This is based on the following interpretation: 3 < BF < 10 can be considered moderate evidence and BF > 10 can be considered strong evidence (*van Doorn et al., 2021*).

## DISCUSSION

The purpose of the current study was to examine whether dog ownership moderates the effects of viewing images of puppies (compared with images of adult dogs) on the performance of two computerized RT-based tasks. We hypothesized that dog owners who viewed images of puppies would perform better than dog owners who viewed images of adult dogs and that this effect would not be found in non-dog owners. Our data did not support our hypotheses as there were no differences in performance after viewing images of either puppies or adult dogs in dog owners.

However, we found that dog owners reacted faster than non-dog owners in the alternate task-switching task after viewing images of dogs. This finding emerged from our pre-registered analysis had a relatively small effect size ($\eta^2_p = 0.03$). This small effect size coincides with small to moderate effect sizes that differentiated between dog owners and non-dog owners in the way they rated cuteness ($\eta^2_p = 0.02$) and pleasantness ($\eta^2_p = 0.06$) (there were no difference between dog owners and non-owners in any of the other attributes). Therefore, we explored this finding with two additional analyses (see Exploratory Analyses section). Out of those analyses, one supported our original finding that dog owners were faster than non-dog owners after viewing images of dogs and one did not support it. In addition, 41 non-dog owners and only four dog owners marked zero-to-three on dog loving scale. These participants had slower RTs compared with the 55 dog owners and 29 non-dog owners who marked five on the dog loving scale. This provides supports to the finding of slower performance in non-dog owners because they are the ones that almost exclusively occupied the zero-to-three group. Based on these data, it could be argued that is dog loving rather than dog ownership that affected performance.

However, because this finding is based on an exploratory analysis, it can only generate this hypothesis for future studies and cannot be concluded from the current study.

Despite the relatively small effect size and the different findings resulting from alternative analyses, the finding that dog owners outperformed non-dog owners is of interest. Indeed, in a previous study (with the same methodology and tasks), we found similar differences between pet owners and non-pet owners (but in the Simon task and not in the alternate task-switching task) (*Ziv & Fox, 2021*). This finding suggests that when dog owners view images of dogs, some underlying processes may lead to improved performance. These mechanisms can include, among other things, increased motivation, or more positive affect in general. Indeed, motivation can have a positive effect on motor performance and learning (*Wulf & Lewthwaite, 2016*) but also in other domains (*e.g.,* reading performance (*Retelsdorf, Köller & Möller, 2011*), academic performance (*Kusurkar et al., 2013*), creative thinking (*Halpin & Halpin, 1973*)). In addition, motivation can affect attention and the performance of attentional-based tasks, and thus, motivational factors should be considered when interpreting performance based on neuropsychological processes (*Robinson et al., 2012*). In the current study, we suggest that showing participants with an affinity for dogs (*i.e.,* dog owners), images of puppies and of adult dogs, can increase their affect, their motivation, and their attention to the tasks. These changes, in turn, may lead to improved performance.

Even though we found the difference between dog owners and non-dog owners only in the post-test, another possible explanation, although speculative, can be offered. The dog owners in the current study may be more active than the non-dog owners. It has been previously shown that dog ownership is associated with more mild-to-moderate physical activity (*Brown & Rhodes, 2006*). *Christian et al. (2013)* reviewed published studies on dog ownership and physical activity between 1990–2010 and found that 60% of dog owners walked their dogs four times per week for a total of 160 min. In this respect, physical activity has been associated with improved executive function. For example, *Kamijo & Takeda (2010)* showed that young participants who were physically active outperformed their sedentary counterparts in a task switching paradigm. *Hillman et al. (2006)* found that physically active participants responded faster than their sedentary counterparts in task switching. In addition, physically active participants also showed faster P3 component with greater amplitudes as measured with electroencephalogram. Differences in the latency of P3 are thought to be related to the speed of cognitive processing and differences in the amplitude of P3 appear to be related to the allocation of attentional resources in updating working memory (*Hillman et al., 2006*). Therefore, the better performance of the dog owners in the task-switching task can be explained, at least in part, by their regular engagement in physical activity.

Other than the abovementioned finding, our study produced only null results that can be interpreted in several ways. First, an intervention in which we show participants images of puppies and adult dogs is not expected to produce large effect size. Indeed, *Funder & Ozer (2019)* suggested that in psychological research, small effects from large sample sizes are likely to represent the actual state of events. Hence, we would also expect modest effect sizes in the current study, and it is also plausible that studies on this topic will produce

variable results. In this respect, the sample size of the current study was calculated based on a moderate effect size and it is possible that smaller effects were not detected. It is also possible, however, that the manipulation in the current study simply did not lead to the hypothesized effect. Even though several previous studies found an effect of viewing cute images on performance (*e.g.*, *Nittono et al., 2012*; *Sherman, Haidt & Coan, 2009*; *Yoshikawa & Masaki, 2021*; *Ziv & Fox, 2021*), it appears that this effect is not robust and does not lead to consistent changes in performance.

Another possible explanation for the null findings is that the tasks were not difficult or sensitive enough to expose the effects of the manipulation. However, these tasks were used in a previous study with the same methodology (*Ziv & Fox, 2021*) and differences between participants who had pets and participants who did not have pets were found. In addition, a study on the effect of providing choice on performance that used the same computerized tasks found differences in RT between participant who were given the option to choose the order of the tasks they performed compared with participants who were not given such choice (*Ziv & Lidor, 2021*). Hence, we believe that the tasks were appropriate.

### Study's strengths and limitations

The current study has two main strengths. First, we were able to use a large sample size that provided us with the necessary statistical power to find moderate effect sizes. Second, using an online participants recruitment service allowed us to blind participants to the study's purpose because: (1) recruitment was covert and participants did not know why they were recruited for the study, and (2) there was no contact between the researchers and the participants and so there was no way that researchers' behavior/affect/tone of voice can influence participants' performance.

However, using an online study format comes with limitations as well. One limitation is that the researcher cannot directly know whether the participants are attentive enough and perform the tasks in a serious manner. To account for participants who are not engaged in the experiment we deleted blocks with 13 correct responses or less (out of 24). In addition, *Woods et al. (2015)* suggested that the larger sample sizes that can be reached in online studies can offset the lack of experimental control and that, compared with online studies, laboratory-based studies do not necessarily allow for more attentive participation.

One final limitation is the duration of the exposure to the treatment. In the current study, participants spent approximately 40–45 s rating the images of the puppies or adult dogs between the pre-test and the post-test. in previous studies (*Nittono et al., 2012*; *Yoshikawa, Nittono & Masaki, 2020*) participants spent 90 s rating the images. It is possible that the time participants spent rating images in the current study was too short to produce the behavioral carefulness effect. In future online studies, longer viewing times for such interventions can be used.

## CONCLUSION

In conclusion, the current study did not show any differences in performance of two computerized RT-based tasks after viewing images of puppies or adult dogs. Therefore, the hypothesized behavioral carefulness was not replicated. However, we found a small effect

showing that dog owners performed faster than non-dog owners in an alternate task-switching task after viewing images of dogs. This finding suggests that the presentation of dogs to dog owners may affect their motivation and attentiveness to the task performed.

### Funding
The authors received no funding for this work.

### Competing Interests
The authors declare that they have no competing interests.

### Author Contributions
- Orly Fox conceived and designed the experiments, performed the experiments, analyzed the data, prepared figures and/or tables, authored or reviewed drafts of the article, and approved the final draft.
- Gal Ziv conceived and designed the experiments, performed the experiments, analyzed the data, prepared figures and/or tables, authored or reviewed drafts of the article, and approved the final draft.

### Human Ethics
The following information was supplied relating to ethical approvals (*i.e.*, approving body and any reference numbers):

The ethics committee of the Academic College at Wingate.

### Data Availability
The raw dataset is available in the Supplemental Files and at OSF: Ziv, Gal. 2022. "Does Dog Ownership Moderate the Behavioral Carefulness Effect." OSF. July 27. osf.io/48azq.

### Supplemental Information
Supplemental information for this article can be found online at http://dx.doi.org/10.7717/peerj.14439#supplemental-information.

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
