# Peer review of "The effects of viewing cute images on the performance of simple computerized tasks in dog owners and non-dog owners"

_PeerJ, doi:10.7717/peerj.14439_

## Round 0.1 · original submission · Major Revisions

Your manuscript has been assessed by two expert reviewers. They both found the study to be important but also had concerns. Based on these reports, and my own assessment as Editor, I think the manuscript has potential but I have decided that major revisions are needed.

Both reviewers note that the tasks chosen may not have been the correct ones for testing this hypothesis so more explanation/justification is needed here. Reviewer 1 notes some lack of clarity with “rating preferred images” and the details surrounding this part of the experiment, as well as the mention of additional variables (e.g., how much they love dogs) which are never returned to. Reviewer 2 also notes the strange decision to not use Test (pre/post) in the ANOVAs since these would directly address the question under consideration, even though there is some thought on this later on terms of the ANCOVA.

In addition, my own concerns mirror this lack of direct comparison between pre- and post-test performance, which perhaps is an issue of power? Whatever the reason, it needs some more explanation. I also feel that more information is needed in certain areas. For instance, on the tasks themselves (though I accept the reader can indeed refer to other work for more info too), the animal images viewed (why seven? viewed for how long?), where the data exclusion criteria came from (and so need justifying), and whether the male/female comparison was powered. (The power analysis reported does not seem to apply to the analyses of switch vs no-switch trials, from what I can tell.) Also, for null effects, these can be hard to demonstrate and many readers might like to see Bayes factors reported. p-values should be reported to three decimal places. And finally, the only significant effect (dog owners reacted faster than non-dog owners) had only a small effect size, I think, and was not predicted so maybe should not be made much of.

Reviewer 1 ·

Basic reporting

This paper reports a single online study in which the effects of viewing baby vs. adult animal images on the performance of a Simon task and an alternate task-switching task. The protocol was pre-registered and the experiment was conducted accordingly with a minor change that did not seem to affect the validity of the study. The paper is generally well-written with adequate literature references. Raw data are shared.

Experimental design

This experiment followed the authors' previous experiment (Ziv & Fox, 2021, Frontiers in Psychology) and used the same tasks. As the authors mentioned in the Discussion, the selection of these tasks may not be optimal for testing the effects of viewing cute mages. However, it is reasonable to use these tasks in this study because the authors' purpose was to replicate their previous findings.

Validity of the findings

Although I don't have any serious doubts on the results, the order of sections in the Results can be reconsidered. For example, I'm not sure what "Time spent rating preferred images" (Line 160) means, in particular the meaning of "preferred." As far as I understand, the participants were asked to rate the images they saw during the task pause, regardless of their preference. The number of images to be rated should also be specified. This section can be reported later, in combination with the rating results. Similarly, the section "Difference between males and females" (Line 168) can be moved to an appropriate place.

I think the rating results in the main study (Lines 231-258) would be better to be shown in a table format, possibly incorporated with Table 1 (preliminary ratings), for better comprehension.

It is useful to mention the differences between dog owners and non-owners more clearly. For instance, only the cuteness and pleasantness ratings differed between the two groups. This finding is worth highlighted in the Discussion and possibly in Table. Moreover, the result of "how much they loved dogs" (Line 137) is not reported. This variable can also be different between owners and non-owners.

Additional comments

Line 125. Please mention how long each task lasted in each block.

Lines 309 and 310. Please specify what "group" means.

Reviewer 2 ·

Basic reporting

no comment

Experimental design

no comment

Validity of the findings

no comment

Additional comments

The authors tried to replicate previous findings of the cute-viewing effect, using a Simon task and a task-switching task in a between-subjects design. Using only dog images, they compared performance between dog owners and non-dog owners. They failed to obtain any cute effect contrary to their hypotheses. However, they found a dog-owner effect that shortened reaction time in the task-switching task after viewing dog images (independent of maturity) only in dog owners.
I found these results important given an insufficiency of empirical evidence for the cute effect. Nevertheless, I would like to raise several questions and comments especially pertaining to their analyses and interpretations of the results. Although I only describe a few problems below, a thorough revision should make this manuscript acceptable for publication.

Introduction:
The authors should clearly explain why they used a Simon task and a task-switching task in the Introduction section. I did not find any convincing reason or grounds for this decision. I suspect that they did not select appropriate tasks to replicate the cute-viewing effect. In previous studies of the cute-viewing effect, the tasks required attentional “focus” that was needed to detect an important property embedded in the stimulus, inducing approach motivation. It is true that both the Simon task and the task-switching task require attention to be executed, but attention is always shifted between the relevant and irrelevant stimulus properties in the Simon task and attention is needed to detect a signal of the rule change in the task-switching task. Previous studies have found positive cute-viewing effects in those tasks that required focal attention, approach motivation, and dexterity. I wondered why the authors did not use a global-local letter task if they wanted to replicate previous findings.

I suspect that the authors misread a previous finding of Nittono et al., 2012 (line 56). Nittono et al. did not record RTs in Experiment 2. Even if the authors referred to results of Experiment 3, Nittono et al. reported a reduction of the global precedence effect probably due to a longer RT for the global dimension of the stimulus. The authors should properly refer to the paper.


Methods:
As for the statistical analysis of switch and no-switch trials in the task-switching task (Line 197), they should include a factor “Test (pre-test/post-test)” in their ANOVA as long as they refer to the difference between pre-test and post-test. They should not have separately analyzed performance in each test. This is needed even though they additionally adopted ANCOVA in the last section.

Results:
Although “no Dog ownership group effect (Line 201)” was reported in this study, mean values are needed to compare results between the pre-test and the post-test.

Table 1 should show values separately for dog owners and non-dog owners.
Figure 1 should show mean RTs associated with the dog ownership separately for the pre-test and the post-test.

Interpretation:
Did the authors record subjects’ characteristics such as exercise habit, frequency of their dog walking, fitness, and BMI? It is well-known that dog owners tend to have a habit of walking with their dogs. In addition, there have been numerous findings that higher fit individuals exhibit better executive function that may result in shorter RT than lower fit individuals. It is possible that the habit of walking may be responsible for the shorter RTs even though RT did not differ between groups in the pre-test. It may also be that dog owners are simply more familiar with dogs and better able to distinguish between dogs, which may have led to the shorter RTs.

---

## Round 0.2 · Major Revisions

Thank you for addressing the reviewers' comments. However, both reviewers continue to have a number of concerns with the manuscript, so I am inviting you to revise your manuscript once again. Be sure to address each and every comment to the best of your abilities.

Reviewer 1 ·

Basic reporting

no comment

Experimental design

no comment

Validity of the findings

I appreciate the authors' effort to revise the paper considerably according to the editor's and reviewers' comments. However, the revision makes me more confused than before. I have two serious concerns.

First, the results of new analyses (a 4-way ANOVA and Bayes Factors) did not support the authors' initial argument that dog owners reacted faster than non-dog owners in the post-test of the alternate task-switching task. Generally, it is not recommend to increase the number of tests because it will inflate false positive rates. A higher-order ANOVA itself has a risk to inflate them (e.g., Luck et al., 2017, https://doi.org/10.1111/psyp.12639). On the other hand, Reviewer 2 is correct in that an ANCOVA is inappropriate to use here. This is because it does not address the group difference between dog owners and non-owners at the pre-test period properly. The use of ANCOVA is warranted when the two groups are randomly assigned to two experimental conditions such as image types (baby vs. adult). Because this is a pre-registered study, I think it is reasonable to stick to the original method. However, the real problem is that BF results did not support the authors' claim. BF01 that the authors reported indicates evidence in favor of the null hypothesis. As far as I understand, no evidence is provided for the alternative hypothesis (BF10) that dog owners reacted faster than non-dog owners.

Second, I don't understand the rationale to analyze sex differences exploratorily. Because there is no remark on this subject in the Introduction, I think it is not appropriate to explore them.

In sum, I have an impression that the authors did not directly answer to their initial hypotheses: (1) In dog owners, in both a Simon and an alternate task-switching task, participants who viewed images of puppies would make more correct responses and react faster compared with participants who viewed images of adult dogs, and (2) in non-dog owners, these effects would not be found (ll. 90-93). As OA journals like PeerJ encourage researchers to report null findings that are obtained by a scientifically valid procedure, I think this paper would be more useful if it answers to the initial questions straightly, regardless of success or failure.

Additional comments

Minor points.

line 158 "to complete for" Delete "for"

Line 194 "used student's t-test" would be "used a Student's t-test" or "used Student's t-tests."

Lines 313-314. "no difference in the time it took participants to look at the seven images" would be "no difference in time the participants spent to look at ... and to rate ..." Please consider which verb is more appropriate, "rate" or "rank" for this procedure.

Reviewer 2 ·

Basic reporting

Justification is still needed.

Experimental design

no comment

Validity of the findings

no comment

Additional comments

I found that the revised manuscript was significantly improved. However, I still have two concerns that should be addressed in a revision. I found a discrepancy between reasons why they used the tasks and their hypothesis. I would like to encourage the authors to justify their premise more clearly. See below for details.

1. The authors provided the following explanation for their selection of the two tasks in the revised manuscript. However, I do not understand the first reason “We chose these tasks because: (1) we wanted to replicate a previous study that used the same tasks (Ziv & Fox, 2021)…”
However, Ziv & Fox (2021) did not find any cuteness-viewing effect; therefore, if they wanted to replicate the findings of Ziv and Fox (2021) they must have expected a null result for the cuteness-viewing effect. Nevertheless, they hypothesized in the next paragraph that “in both a Simon and an alternate task-switching task, participants who viewed images of puppies would make more correct responses and react faster compared with participants who viewed images of adult dogs.” I was confused by this discrepancy. They should make their descriptions more coherent.
In addition, it is still difficult to follow their logic associated with the speed-accuracy trade-off. Indeed, Nittono et al. found a speed-accuracy trade-off (i.e., a more careful execution after viewing cute pictures). If the authors wanted to replicate previous findings, I do not understand why they used RT tasks in which both speed and accuracy are emphasized. Perhaps tasks unrelated to such emphasis might be suitable to test the cuteness-viewing effect. I think more corrections are needed to justify their premise.

2. I did not clearly find any answer to my question about subjects’ characteristics in the authors’ response. I assume that perhaps they were not recorded prior to the experiment.
The authors cited a paper that reported the beneficial effects of exercise on executive functions. There have been several studies that tested the topics in terms of chronic effects (e.g., Hillman et al., 2006; Kamijo et al., 2010) and acute effects (Tomporowski, & Ganio, 2006, Int. J. Sport Exerc. Psychol.; Bae & Masaki, 2019, Front. Hum. Neurosci.) using the task-switching paradigm. I believe that it is fairer and more appropriate to cite Hillman et al.’s paper (International Journal of Psychophysiology 59 (2006) 30 – 39) instead of Kamijo et al (2010), if they choose one of these studies.

---

## Round 0.3 · Minor Revisions

Thank you for your work on this manuscript. As you can see, there are just a couple of comments from Reviewer 1 that need addressing before your article can be accepted for publication.

Reviewer 1 ·

Basic reporting

The authors have answered my concerns clearly. I have only two further comments.

Line 87. "in addition" Capitalize the first letter.

Line 288. "The Switch condition alone was less likely (BF01 = 1.5)" "condition" means "factor"? I'm curious why the Switch factor was associated with a BF that supports the null hypothesis (BF01 = 1.5), given that the Switch factor had a large effect size in the ANOVA. Also, BF10 for the model "Switch + Dog-owner" is extremely high, 2.13E+37. Please check these values again.

Experimental design

No comment.

Validity of the findings

No comment.

Additional comments

no comment.

Reviewer 2 ·

Basic reporting

I appreciate that the authors have successfully addressed all my concerns. The revised MS now presents a convincing argument and is suitable for publication.

Experimental design

no comment

Validity of the findings

no comment

---

## Round 0.4 · accepted · Accept

Thank you for your hard work on this manuscript, and for addressing the reviewers' comments satisfactorily. I look forward to seeing the article when it's published.